# Quality or Quantity? How Structural Parameters Affect Catalytic Activity of Iron Oxides for CO Oxidation

Steffen Schlicher [1], Nils Prinz [2], Julius Bürger [3], Andreas Omlor [4], Christian Singer [5], Mirijam Zobel [2], Roland Schoch [1], Jörg K. N. Lindner [3], Volker Schünemann [4], Sven Kureti [5] and Matthias Bauer [1,*]

1   Department of Chemistry and Center for Sustainable Systems Design (CSSD), Paderborn University, Warburger Straße 100, 33098 Paderborn, Germany; steffen.schlicher@upb.de (S.S.); roland.schoch@upb.de (R.S.)
2   Institute of Crystallography, RWTH Aachen University, Jägerstr. 17-19, 52066 Aachen, Germany; prinz@ifk.rwth-aachen.de (N.P.); zobel@ifk.rwth-aachen.de (M.Z.)
3   Department of Physics, Paderborn University, Warburger Straße 100, 33098 Paderborn, Germany; buergerj@mail.uni-paderborn.de (J.B.); lindner@physik.upb.de (J.K.N.L.)
4   Department of Physics, Technical University of Kaiserslautern, Erwin-Schrödinger-Straße 46, 67663 Kaiserslautern, Germany; omlor@rhrk.uni-kl.de (A.O.); schuene@physik.uni-kl.de (V.S.)
5   Institute of Energy Process Engineering and Chemical Engineering, Chair of Reaction Engineering, Technical University of Freiberg, Fuchsmühlenweg 9, 09599 Freiberg, Germany; christian.singer@iec.tu-freiberg.de (C.S.); sven.kureti@iec.tu-freiberg.de (S.K.)
*   Correspondence: matthias.bauer@upb.de

**Abstract:** The replacement of noble metal catalysts by abundant iron as an active compound in CO oxidation is of ecologic and economic interest. However, improvement of their catalytic performance to the same level as state-of-the-art noble metal catalysts requires an in depth understanding of their working principle on an atomic level. As a contribution to this aim, a series of iron oxide catalysts with varying Fe loadings from 1 to 20 wt% immobilized on a γ-$Al_2O_3$ support is presented here, and a multidimensional structure–activity correlation is established. The CO oxidation activity is correlated to structural details obtained by various spectroscopic, diffraction, and microscopic methods, such as PXRD, PDF analysis, DRUVS, Mössbauer spectroscopy, STEM-EDX, and XAS. Low Fe loadings lead to less agglomerated but high percentual amounts of isolated, tetrahedrally coordinated iron oxide species, while the absolute amount of isolated species reaches its maximum at high Fe loadings. Consequently, the highest CO oxidation activity in terms of turnover frequencies can be correlated to small, finely dispersed iron oxide species with a large amount of tetrahedrally oxygen coordinated iron sites, while the overall amount of isolated iron oxide species correlates with a lower light-off temperature.

**Keywords:** CO oxidation; iron oxide; emission control; PDF; STEM-EDX mapping; XAS

## 1. Introduction

Emission reduction nowadays is of great importance, since the world population and hence the pollution of air, soil, and water is steadily growing [1–4]. A well-established method to keep emission levels below certain thresholds is to convert pollutants into their harmless derivatives, often by the utilization of a catalyst. Prominent examples are the photocatalytic treatment of wastewater [5–7], the removal of particulate matter [8,9], or the selective catalytic reduction (SCR) of nitric oxides to elemental nitrogen [10–13]. Modern combustion engines are equipped with catalytic converters in which, besides the reduction of $NO_x$ and the oxidation of hydrocarbons, CO is oxidized by residuary oxygen to non-toxic $CO_2$ [14–17]. So-called three-way catalysts or diesel oxidation catalysts, as an example, require the use of noble metals, namely rhodium, palladium, and platinum. Due to both economic [18] and ecological [19] reasons, the search for alternatives to these noble metals has gained more and more attention. Various abundant transition metals, such as cobalt

or copper, have been investigated regarding their ability to convert CO to $CO_2$ but none of these have been of reasonable relevance for industrial applications yet [20–30]. Iron has been tested by various groups in this context since the 1960s but cannot yet compete with the catalytic activity of state-of-the-art catalysts [31–33]. During the last two decades, the demand for sustainable alternatives has grown, as has the interest in iron oxides for catalytic CO oxidation. Mentionable approaches include the doping of quartz wool with iron oxides by Li et al. [34] or a precipitation method for high-surface-area iron oxide catalysts with very small particle sizes [35]. Different support materials were tested, such as mesoporous zeolites [36] or even minerals such as bentonite [37]. Despite a manifold of different approaches [38–42], iron catalysts are still not understood in detail to challenge noble metal catalysts in terms of activity and stability. Schoch et al. showed a facile route for the preparation of iron oxide catalysts supported on $\gamma$-$Al_2O_3$ with excellent catalytic activity in CO oxidation experiments [43]. They also ascertained differences in the catalyst activity with structural variations. High amounts of tetrahedrally coordinated iron oxide species tended to have higher CO oxidation activity than catalysts with $Fe^{3+}$ in octahedral coordination geometry or even bulk phase.

Here, we aim to contribute to the application of iron oxides for CO oxidation under lean conditions based on a systematic structure–activity correlation. For this purpose, five iron oxide catalysts with Fe loads from 1 to 20 wt% on a $\gamma$-$Al_2O_3$ support were prepared by wetness impregnation with $Fe(acac)_3$ and investigated in catalytic CO oxidation reactions. Extensive characterization by a broad range of analytical tools was applied, such as powder X-ray diffraction (PXRD, HR = high resolution), pair distribution function (PDF) analysis, diffuse reflectance UV/Vis spectroscopy (DRUVS), Mössbauer spectroscopy, scanning transmission electron microscopy coupled with energy-dispersive X-ray spectroscopy (STEM-EDX), and X-ray absorption spectroscopy (XAS). Thereby, we wish to elucidate the structural parameters of iron oxide catalysts that promote high catalytic activity, especially at low temperatures and by means of high rates of conversion. These findings can then be used for the development of new synthetic routes to prepare catalysts that could eventually show extraordinary catalytic activities, ready to compete with platinum group metals.

## 2. Results

### 2.1. Structural Characterization

The nomenclature of the catalysts in the following discussion is FeX, with X reflecting the nominal loading of iron on the powdered $\gamma$-$Al_2O_3$. The surface area can already give important information about catalytic systems. The pure $\gamma$-$Al_2O_3$ exhibits a BET surface area of 169 $m^2$/g, which does not change significantly for Fe01 to Fe10 (Table 1). For Fe20, the area decreases to 121 $m^2$/g, which indicates the agglomeration of the iron species and, as a result, reduced accessibility of the metal centers.

**Table 1.** Specific surface areas of Fe01 to Fe20 compared to the pure $\gamma$-$Al_2O_3$ support, obtained via BET method.

| Sample | BET Surface [$m^2$/g] |
|---|---|
| $\gamma$-$Al_2O_3$ | 169 |
| Fe01 | 155 |
| Fe025 | 164 |
| Fe05 | 167 |
| Fe10 | 152 |
| Fe20 | 121 |

The combination of HRPXRD and subsequent PDF analysis is highly suited to investigate the long- and medium-range order of nanoparticular species, such as the crystalline and nanostructured phases of the support material as well as the catalytic species [44–46]. For the highly loaded samples Fe10 and Fe20, the HRPXRD patterns (Figure 1) indicate (besides reflexes of the support) reflexes of both $\alpha$-$Fe_2O_3$ and $\gamma$-$Fe_2O_3$.

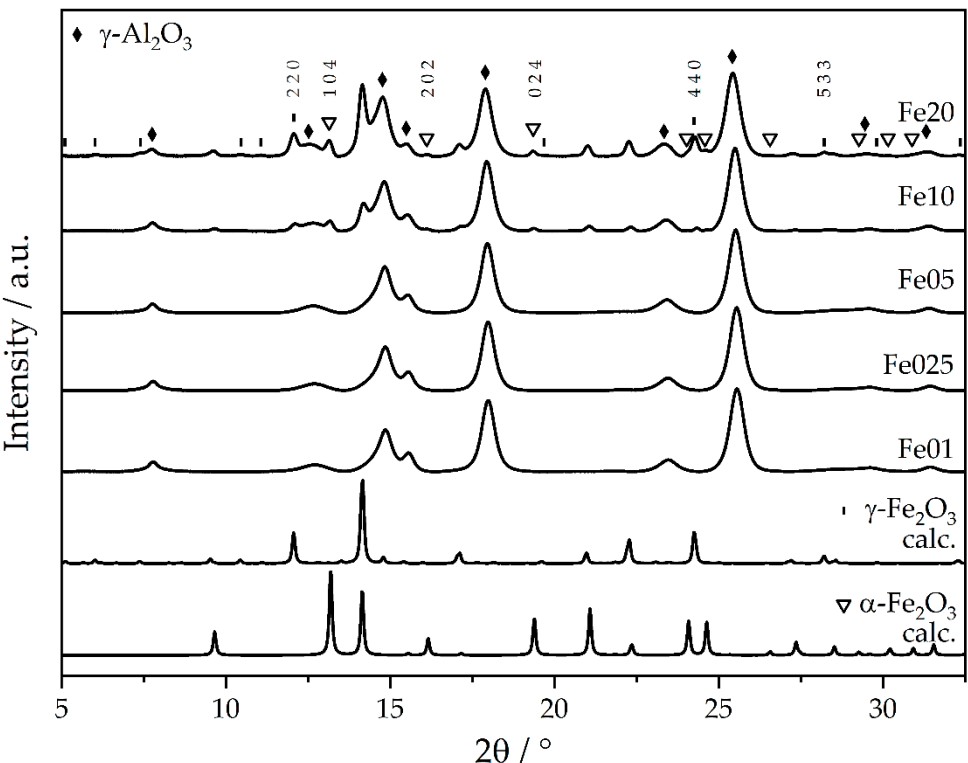

**Figure 1.** Experimental high-resolution powder X-ray diffractograms of catalysts Fe01 to Fe20 and calculated diffractograms of $\gamma$-Fe$_2$O$_3$ and $\alpha$-Fe$_2$O$_3$, respectively, maghemite and hematite (beamline P24, DESY).

Only the reflexes of the lattice planes (104), (202), and (024) of $\alpha$-Fe$_2$O$_3$ [47], as well as the lattice planes (220), (440), and (533) of $\gamma$-Fe$_2$O$_3$ [48], can clearly be indexed. For lower Fe loadings, no iron oxide reflexes are observed. However, the Bragg reflexes of $\gamma$-Al$_2$O$_3$ progressively shift to smaller 2θ values for increasing iron loads (Figure 2). This shift corresponds to an increase in the lattice parameters, which is isotropic due to its cubic crystal class. Exact Bragg angles of the $\gamma$-Al$_2$O$_3$ crystal lattices (311), (222), (400), (511), and (440) were retrieved from pseudo-Voigt fits of the reflexes [49,50]—all five reflexes shift in a similar manner. The (111) and (220) peaks could not be analyzed this way, due to overlap with iron oxide peaks for higher loadings. This change in the lattice spacings of the support material amounts to 0.03–0.60% of the bulk $\gamma$-Al$_2$O$_3$ lattice parameters for the different loadings and points towards the restructuring of the $\gamma$-Al$_2$O$_3$ substrate. Such a restructuring could be explained by the incorporation of Fe ions into the inverse spinel structure of $\gamma$-Al$_2$O$_3$ and hence expansion and local distortions [51]. Since the $\gamma$-Al$_2$O$_3$ support is porous (see BET results), much surface is available to interact with the Fe during impregnation and calcination. Further, the interface between Fe$_2$O$_3$ particles and the $\gamma$-Al$_2$O$_3$ support can bear a locally different structure than the bare support due to restructuring of the $\gamma$-Al$_2$O$_3$ in the presence of the Fe$_2$O$_3$ without explicit Fe incorporation into the $\gamma$-Al$_2$O$_3$. The majority of the $\gamma$-Al$_2$O$_3$ remains unaltered, as deduced from the overlapping $\gamma$-Al$_2$O$_3$ peak areas in Figure S3 top.

In order to gain insights into the local structure of the iron oxide entities on the support, PDF analysis was carried out. First, the structure of the bare $\gamma$-Al$_2$O$_3$ support was refined against a cubic (inverse spinel) Fd$\bar{3}$m crystal structure, with Al$^{3+}$ cations on tetrahedral and octahedral spinel positions, as well as additional Al$^{3+}$ cations on non-spinel positions. The experimental PDF is described well over a fit range of 1–80 Å, with a goodness-of-fit $R_w$ = 0.24 (see Figure S4 top). In particular, at r < 10 Å, some structural features remain in the difference curve, pointing towards a modified short-range order, as previously described in the literature [52]. This modified short-range order involves mainly an underestimation

of Al-Al interatomic distances at 2.82 and 3.31 Å, which represent interatomic distances between octahedral–tetrahedral and octahedral–octahedral positions within the spinel structure, respectively (see Figure S4 bottom). The match between experimental and theoretical PDF can be significantly improved to a $R_w$ of 0.20 (see Figure S5), when refining only the 1–10 Å range with a different occupancy for certain $Al^{3+}$ positions (see Table S1 for refinement values). Some deviations still remain—for instance, the discrepancy of the O-O interatomic distance at approx. 4.9 Å—which could be explained by aperiodic stacking faults in the oxygen sublattice [52].

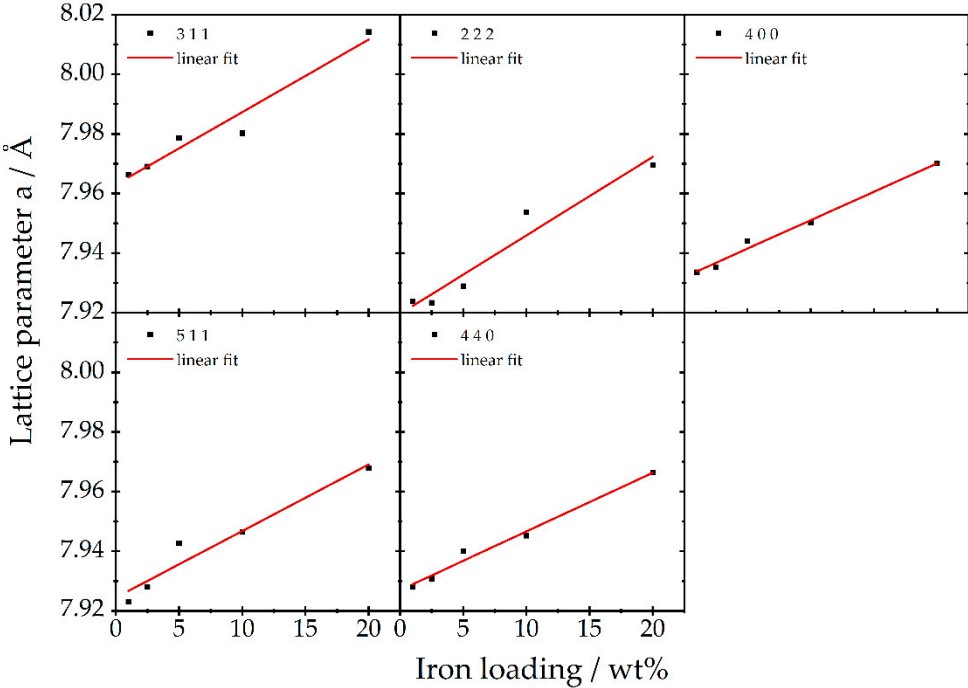

**Figure 2.** Cubic lattice parameter *a* obtained for different crystal planes of the $\gamma$-$Al_2O_3$ support of Fe01 to Fe20 and the corresponding linear regression, as determined from high-resolution XRD data (beamline P24, DESY).

Then, the catalysts were refined with multiphase fits (see Figure 3 for Fe20 with the experimental data in blue and the refinement in red). The fit contains a $\gamma$-$Al_2O_3$ phase (grey) and a $\gamma$-$Fe_2O_3$ phase (orange). For both phases, the cubic lattice parameters $a(\gamma$-$Fe_2O_3)$ and $a(\gamma$-$Al_2O_3)$ were refined using $Al^{3+}$ occupancies obtained from the fit of the bare support, as well as the isotropic atomic displacement parameter and a spherical particle size for the iron oxide phase. No $\alpha$-$Fe_2O_3$ phase was considered in the fit, as the PDFs of $\alpha$-$Fe_2O_3$ and $\gamma$-$Fe_2O_3$ are very similar. Refinement with both phases led to high correlation between the refined values and possibly over-parameterization. The lattice parameter of the $\gamma$-$Al_2O_3$ changes from the unloaded to the 20 wt% Fe-loaded sample from 7.919 to 7.962 Å, corroborating the XRD findings of the expanding $\gamma$-$Al_2O_3$ lattice upon iron loading. Again, as for the bare support, the 1–10 Å range shows higher residuals in the difference curve (green). These residuals might stem from the $\gamma$-$Al_2O_3$ support (compare difference curve Figure S4) or very small iron oxide clusters, as the difference curve shows similarities with the short-range order of $\gamma$-$Fe_2O_3$ (orange curve).

Results of similar refinements for the 5 and 10 wt% loadings are summarized in Table 2 together with the data for Fe20. It shows the continuous increase in the $\gamma$-$Al_2O_3$ lattice parameter *a* for higher loadings (in agreement with XRD peak shifts). The spherical $\gamma$-$Fe_2O_3$ particle diameter grows with loading from 2.2 to 8.5 nm, with a sharp size increase between 10 and 20% loading. One might speculate as to whether the range between these two loadings presents a transition range between smaller and larger particles. The lattice parameter of $\gamma$-$Fe_2O_3$ is very small for the 5 and 10% loading (literature bulk 8.347 Å). This

could be caused by internal stress in the very small particles, or by a strong interaction of the particles with the $\gamma$-Al$_2$O$_3$ support, inducing strain or restructuring in the particles.

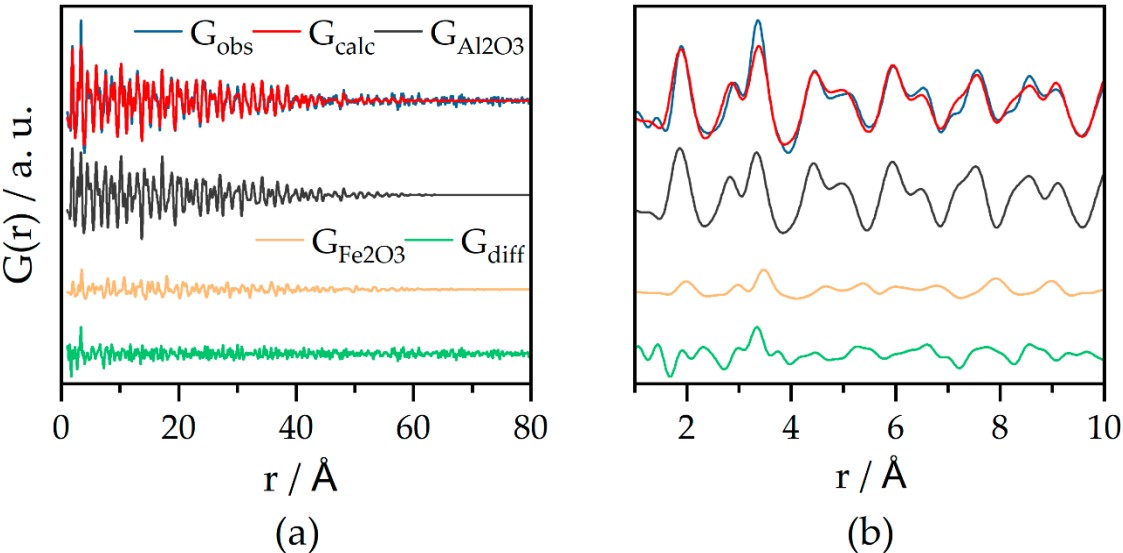

**Figure 3.** (**a**) PDF refinement of laboratory PDF data of Fe20 with experimental data (G$_{obs}$) fitted curve (G$_{calc}$) and difference between G$_{obs}$ and G$_{calc}$ (G$_{diff}$). $\gamma$-Al$_2$O$_3$ (grey) and $\gamma$-Fe$_2$O$_3$ (orange) crystal structures were used; (**b**) magnification of the range of 1–10 Å showing some structural features left in the difference curve; R$_w$ = 0.31; G(r) = reduced pair distribution function.

**Table 2.** Refined values for the lattice parameter of $\gamma$-Al$_2$O$_3$ and Fe$_2$O$_3$ and the spherical particle size of Fe$_2$O$_3$ for Fe05, Fe10, and Fe20.

| Refined Value | Fe05 | Fe10 | Fe20 |
|---|---|---|---|
| $a$($\gamma$-Al$_2$O$_3$) [Å] | 7.935 | 7.949 | 7.962 |
| $a$($\gamma$-Fe$_2$O$_3$) [Å] | 8.025 | 8.062 | 8.327 |
| Spherical particle size (Fe$_2$O$_3$) [nm] | 2.2 | 2.9 | 8.5 |

Since the two-phase PDF refinements leave a significant peak at ca. 3.3 Å in the difference curve and three phases over-parametrize the data, another approach to access the structure of the iron oxide species was carried out. Since the $\gamma$-Al$_2$O$_3$ Bragg peaks are offset to smaller Q, respectively 2$\theta$ (see above), we introduce a compressing factor s = Q$_{440,cat}$/Q$_{440,support}$, calculated from the positions of the Bragg peak maxima of the catalyst-loaded support Q$_{440,cat}$ and the unloaded support Q$_{440,support}$. The XRD pattern of the unloaded catalyst is multiplied with this compressing factor to better subtract the $\gamma$-Al$_2$O$_3$ contributions during d-PDF calculation (see Figure S3 bottom). The resulting d-PDFs mainly contain contributions from iron species (Figure 4), alongside some possibly non-subtracted residuals of $\gamma$-Al$_2$O$_3$, such as local restructuring, which cannot be mimicked by merely accounting for the expansion of the lattice. d-PDFs of Fe05 and Fe10 could only be refined in the range up to 10 Å due to noise in the data. For Fe05 and Fe10, the theoretical Fe-Fe interatomic distance at 2.9 Å is not pronounced well. This could be due to some distortion and overlapping of the two Fe-Fe peaks at 2.9 and 3.3 Å, or due to slightly faulty subtraction of the "compressed" $\gamma$-Al$_2$O$_3$ support. The particle sizes for the 1–10 Å regime are refined to 2.1, 2.3, and 2.8 nm for Fe05, Fe10, and Fe20, respectively, suggesting that very small particles exist. The R$_w$ values of 0.51 and 0.50 for Fe05 and Fe10, respectively, are due to the high noise level in the data.

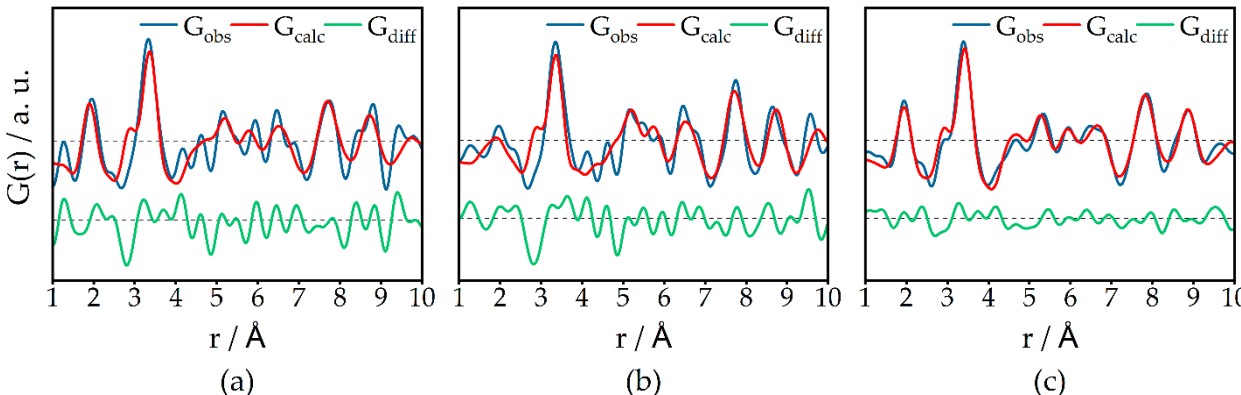

**Figure 4.** d-PDF refinements of Fe05 (**a**), Fe10 (**b**), and Fe20 (**c**) in the 1–10 Å regime using laboratory PDF data. Dashed lines indicate the zero line of the respective refinement. Higher loadings show less noise. The two highest peaks for the Fe-O and Fe-Fe interatomic distances at 1.95 and 3.41 Å are quite well resolved for every loading; G(r) = reduced pair distribution function; $R_w$(Fe05) = 0.51; $R_w$(Fe10) = 0.50; $R_w$(Fe20) = 0.26.

For Fe20, a refinement over 80 Å could be carried out, to elucidate the particle sizes in more detail from the decay of G(r) in the PDF reflecting the particle shape function. Fitting a single $\gamma$-Fe$_2$O$_3$ phase with the particle size as a parameter yielded a poor $R_w$ of 0.60 (see Figure 5a) and fitting a lognormal particle size distribution (fit not shown here) did not significantly improve the fit. Figure 5a shows characteristic peaks in the difference curve from 1 to 7 Å, which hints towards possibly small clusters. Hence, we added the PDF of 1-nm-sized clusters (non-relaxed $\gamma$-Fe$_2$O$_3$ structure cut-out) to the refinement; see Figure 5b. The $R_w$ value significantly improved and peaks at short distances fit better. This small cluster phase has a molar content of 78%, while the larger particle size fraction features a diameter of 14.0 nm. For Fe05 and Fe10, d-PDF refinements cannot detect any $\gamma$-Fe$_2$O$_3$ for r > 10 Å due to noise and/or the absence of larger particles.

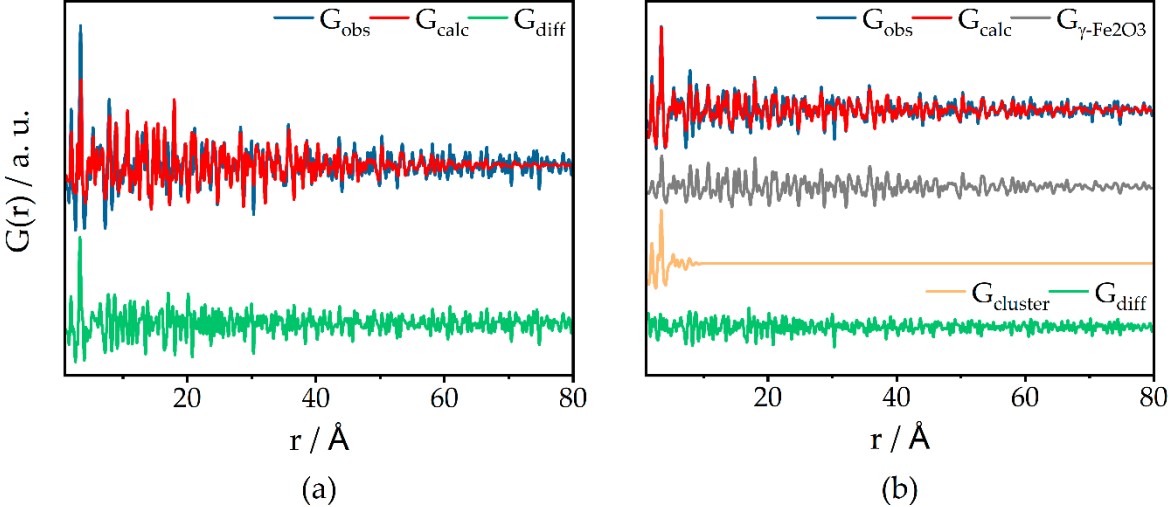

**Figure 5.** d-PDF refinement of Fe20 with a single $\gamma$-Fe$_2$O$_3$ phase (**a**) and an additional 1 nm cluster $\gamma$-Fe$_2$O$_3$ phase (**b**); $R_w$ (**a**) = 0.60; $R_w$ (**b**) = 0.48.

To gain further insights into the cluster sizes of the iron oxide species, additional Mössbauer spectroscopic measurements of catalysts Fe01 to Fe20 were carried out at ambient temperature (298 K) as well as at low temperature (77 K). Due to the small iron concentration in Fe01, no Mössbauer spectrum at low temperature was recorded here. Isomer shifts $\delta$ of Fe025 to Fe20 together with their quadrupole splitting $\Delta E_Q$ verify that

solely iron species in the oxidation state +3 are present in the investigated catalysts (see Figure 6 and the corresponding Table 3). At ambient temperature, all of them show a doublet with an isomer shift of 0.26 to 0.29 mm/s, and since a sextet is missing for Fe01 to Fe05, this can be assigned to isolated iron oxidic species and particles below 13.5 nanometers [53], which is in agreement with the PDF results.

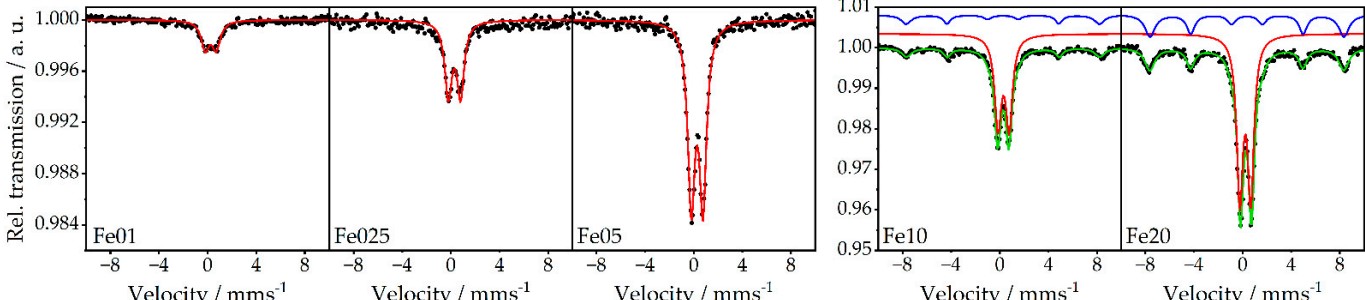

**Figure 6.** Room-temperature $^{57}$Fe Mössbauer spectra of Fe01 to Fe20 (black dots) with the respective fits; red: doublet; blue: sextet; green: cumulative fit.

**Table 3.** Parameters obtained from room-temperature $^{57}$Fe Mössbauer spectra of Fe01 to Fe20. $\delta$ = isomer shift; $\Delta E_Q$ = quadrupole splitting; $\Gamma$ = line width; $B_{hf}$ = magnetic hyperfine field.

| Sample | | $\delta$ [mm/s] | $\Delta E_Q$ [mm/s] | $\Gamma$ [mm/s] | $B_{hf}$ [T] |
|---|---|---|---|---|---|
| Fe01 | | 0.26 | 0.90 | 0.80 | |
| Fe025 | | 0.29 | 1.01 | 0.70 | |
| Fe05 | | 0.28 | 0.96 | 0.70 | |
| Fe10 | doublet | 0.28 | 0.92 | 0.66 | |
| | sextet | 0.28 | 0 | 0.74/0.69/1.0 | 49.3 |
| Fe20 | doublet | 0.27 | 0.90 | 0.61 | |
| | sextet | 0.35 | 0 | 0.79/0.78/1.0 | 49.5 |

The fact that, even at 77 K, no additional sextet occurs for Fe025 and Fe05 indicates that the particles feature superparamagnetism even at 77 K (see Figures S6 and S7). In contrast, Fe10 and Fe20 show additional Zeeman lines due to magnetic splitting even at ambient temperature, which also indicates the presence of iron oxide particles with sizes above 13.5 nm. Areas of the fitted spectra can be used to estimate the ratio of small to large particles. The percentage of doublet compared to sextet changes from 77% at catalyst Fe10 to 69% at Fe20, which means that the fraction of iron species with detectable magnetic coupling, respectively present in particles above 13.5 nm, is much higher for the sample with a higher Fe load.

To elucidate the distribution of iron oxide species in terms of tetrahedrally and octahedrally coordinated $Fe^{3+}$ ions and isolated vs. agglomerated iron oxide species, DRUVS spectroscopy was applied. The local geometry of $Fe^{3+}$ has a strong influence on the absorption bands in the UV/Vis region originating from ligand-to-metal charge transfer (LMCT) transitions. Henceforth, the term oligomers will be used for small agglomerates and the term particles for large agglomerates. Bands below approx. 300 nm can be assigned to isolated iron species, while LMCT transitions of tetrahedral coordinated iron species tend to occur at lower wavelengths compared to those of octahedral coordinated metal centers [54–60]. Oligomers and particles lead to additional bands at higher wavelengths up to 900 nm (for further information, see SI). The spectra of the presented catalysts (Figure 7) show obvious variations in these regions, with Fe01 having the smallest amount of contributions above 400 nm and a main feature at low wavelengths around 300 nm. This feature becomes more intense and shifts to the red with increasing iron loading. Additionally, there is an increase in contributions above 400 nm from Fe01 to Fe20. These observations can be translated into a quantification of the present structures. In $\alpha$-$Fe_2O_3$, solely $FeO_6$

octahedrons are present, while the $\gamma$-$Fe_2O_3$ structure consists of both octahedral $FeO_6$ and tetrahedral $FeO_4$ sites, which means that the ratio of tetrahedrons to octahedrons is a good indicator for the actual iron oxidic phase. Deconvolution of the absorption spectra, as also shown in Figure 8, yields the quantitative information summarized in Table 4. As can be seen in Figure 8a, the normalized area for tetrahedral coordinated isolated $Fe^{3+}$ species decreases from Fe01 to Fe20. The same behavior is found for the area of isolated octahedrons, while the areas for small oligomers and especially for particles increase with increasing iron loading, which is in full agreement with the PDF and Mössbauer results. For Fe01, 12% of the iron centers are found in tetrahedral and 37.3% in octahedral sites, while the rest is present in oligomeric species.

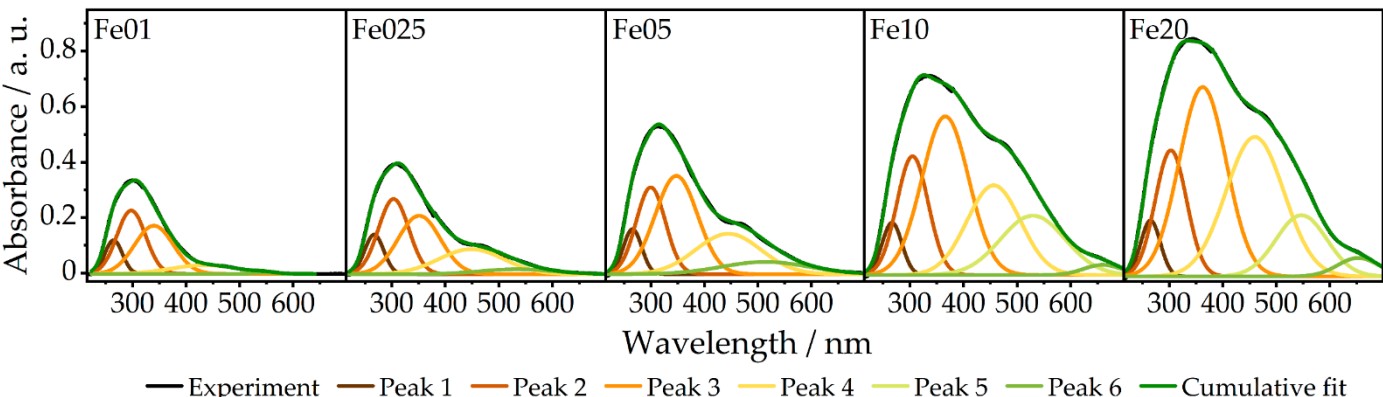

**Figure 7.** Experimental DRUV spectra of Fe01 to Fe20 and the corresponding peak deconvolution with cumulative peak fit.

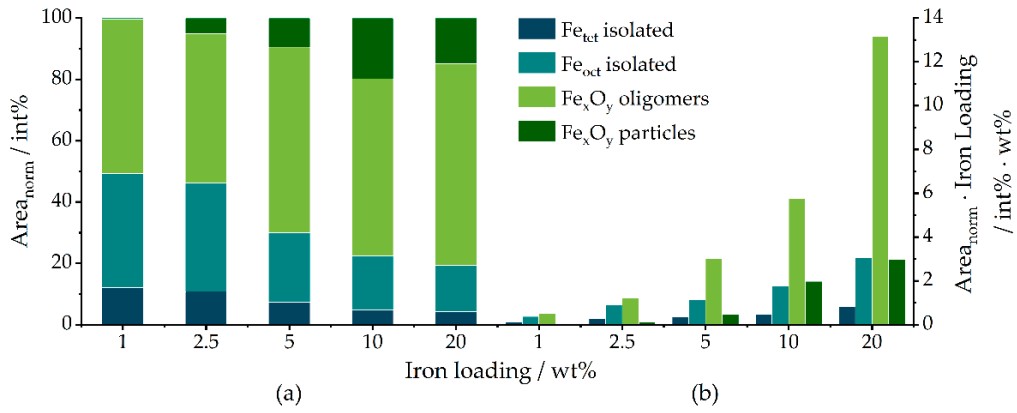

**Figure 8.** (**a**) Normalized areas obtained from peak deconvolution of the DRUV spectra of Fe01 to Fe20; (**b**) total amounts of the species respective to the overall catalyst mass, calculated by multiplication of the integrated percentual areas with the iron loading.

**Table 4.** Parameters obtained from peak deconvolution area of the DRUV spectra of Fe01 to Fe20.

| Catalyst | $Fe_{tet}$ [%] | $Fe_{oct}$ [%] | $Fe_xO_y$ Olig. [%] | $Fe_xO_y$ Particles [%] |
|----------|------------|------------|-----------------|---------------------|
| Fe01 | 12.0 | 37.3 | 50.3 | 0.44 |
| Fe025 | 10.8 | 35.4 | 48.6 | 5.15 |
| Fe05 | 7.30 | 22.7 | 60.5 | 9.52 |
| Fe10 | 4.87 | 17.6 | 57.7 | 19.8 |
| Fe20 | 4.20 | 15.2 | 65.7 | 14.9 |

The fraction of tetrahedral sites decreases constantly with increasing iron loading down to 4.2% for Fe20. Similarly, the structural contribution of octahedral sites decreases to 15.2% for Fe20. Since the crystal structure of $\alpha$-$Fe_2O_3$ does not comprise tetrahedral

coordinated iron oxide species, this increase in the ratio of peak 2 to 1 in the 200 to 300 nm regime at higher iron loadings can be correlated to a decrease in the amount of $\gamma$-$Fe_2O_3$ compared to $\alpha$-$Fe_2O_3$. In the same direction, the number of iron sites in oligomeric and particulate structures increases to maximum values of 66% and 15%, respectively. When the iron loading of each catalyst is taken into account (Figure 8b), the absolute amount of both isolated tetrahedral and isolated octahedral species increases from Fe01 to Fe20, accompanied by a similar increase in oligomers and bulk species. However, due to the number of variables, these findings have to be seen only as an approximation of the actual composition of the investigated catalysts.

To gain insights into the distribution of iron on the support and the iron oxide particle size, scanning transmission electron microscopy of Fe01 to Fe20 was carried out, combined with energy dispersive X-ray mapping. Figure 9 shows the high angle annular dark-field (HAADF) images of each catalyst, together with the elemental maps for oxygen, aluminum, and iron. The catalyst support particles have a rough, irregular appearance with a broad particle size distribution. The HAADF image of catalyst Fe20 shows bright spots with diameters up to 50 nm, corresponding to iron species in the form of small oligomeric iron oxide and particles. The signals of oxygen, aluminum, and iron on the remaining parts are finely distributed. For Fe10, a smaller number of isolated iron oxide agglomerates is found. For Fe05, there were almost no visible particles, and at Fe025, only small particles with a maximum diameter of 7 nm can be found, while Fe01 comprises no visible agglomerates at all. Overall, the observed domains of catalysts Fe01 and Fe025 show a very homogeneous distribution of oxygen, aluminum, and iron signals.

As an extension to the results obtained by UV/Vis spectroscopy, X-ray absorption spectroscopy (XAS) provides a tool to probe the oxidation state and local geometry of the iron centers in the catalyst [61–63]. The iron K-edge X-ray absorption near edge structure (XANES) spectra shown in Figure 10 can be divided into two main features, the prepeak (around 7110 eV to 7120 eV) and the main edge (around 7125 eV to 7135 eV). The K-edge prepeak originates from the excitation of a 1s electron to localized 3d/4p hybrid orbitals. It is therefore sensitive to the symmetry of the probed atom and its oxidation state. The prepeak energies and intensities of all samples are similar and found between 7113.9 and 7114.1 eV (see Table S30). In comparison to this, the prepeaks of $\alpha$-$Fe_2O_3$ and $\gamma$-$Fe_2O_3$ (here represented by the first of two applied fit functions; see Figures S26 and S28) have slightly higher energies, with 7114.6 eV, respectively 7114.3 eV, and with a less broadened and more intense shape.

The main edge shows transitions of 1 s electrons into the continuum, making it a descriptor for the oxidation state of the probed species. The edge positions of Fe01 to Fe20 range between 7125.4 and 7126 eV (see Table S30), while $\alpha$-$Fe_2O_3$ and $\gamma$-$Fe_2O_3$ show edge energies of 7125.1 eV and 7125.9 eV. The prepeak shifts to lower energies of Fe01 to Fe20 are very small, compared to the references, and can therefore be neglected. All obtained energies are in good agreement with the literature and confirm, together with the Mössbauer data, the presence of iron in the oxidation state of +3.

A more detailed view of the local short-range structure of the iron centers can be achieved by analysis of the extended X-ray absorption fine structure (EXAFS) spectra. By application of a model structure obtained from the literature [64], two main coordination spheres can be fitted into the Fourier transformation of the experimental spectra (Figure 11). The first region, from 1.8 to 2.2 Å, containing the oxygen neighbors, allows a discrimination of tetrahedral and octahedral coordination sites. Tetrahedrally coordinated oxygen neighbors feature Fe-O distances below 1.90 Å, while the distances in octahedrons are higher, with 1.96 and 2.13 Å. The region from 2.2 to 4.0 Å contains the nearest iron and aluminum backscatterers. Fitting the experimental data with the EXAFS equation yields the structural parameters summarized in Table 5.

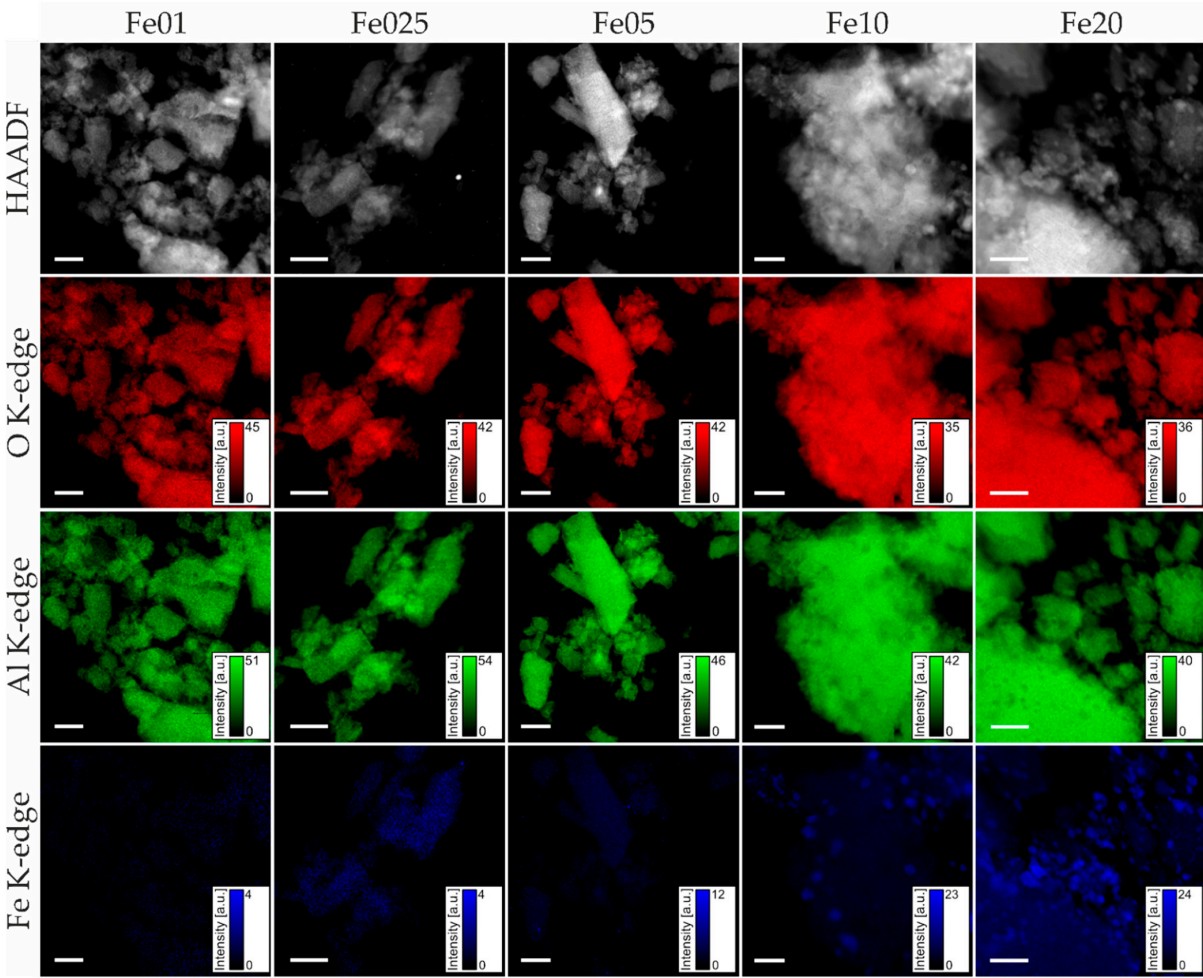

**Figure 9.** HAADF images (top row) and STEM-EDX images (red: oxygen K-edge signal; green: aluminum K-edge signal; blue: iron K-edge signal) of Fe01 to Fe20 (left to right); all scale bars 200 nm.

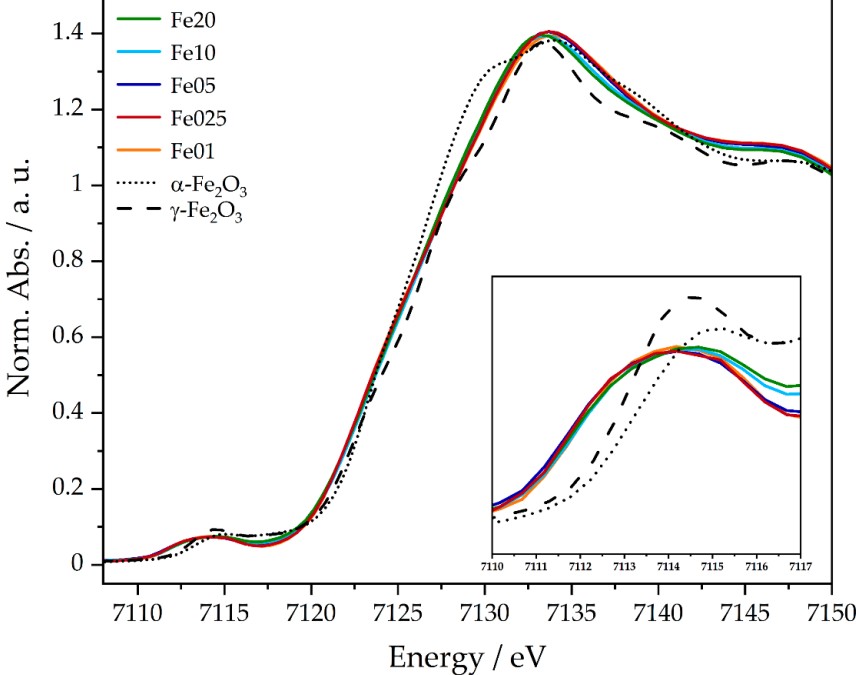

**Figure 10.** XANES spectra of Fe01 to Fe20 and the respective references α- and γ-Fe$_2$O$_3$ at the Fe K-edge.

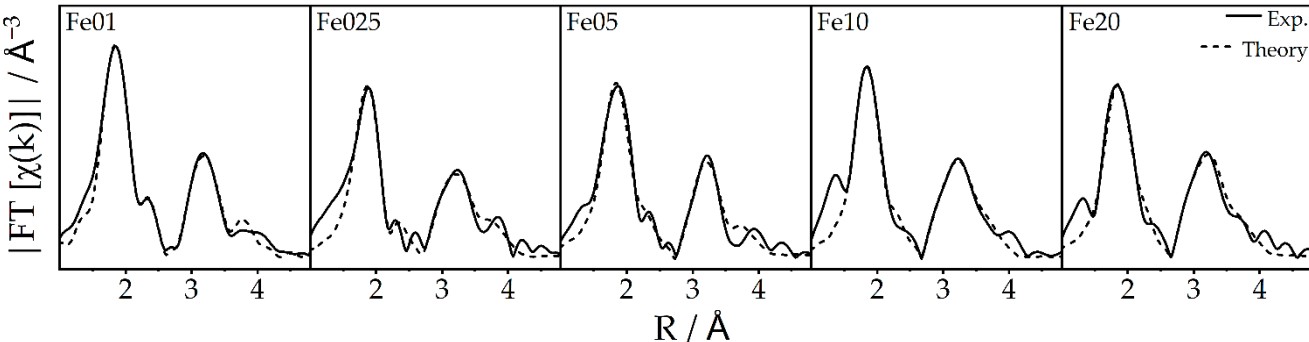

**Figure 11.** Fourier-transformed EXAFS spectra of Fe01 to Fe20 and the corresponding fits.

**Table 5.** Structural parameters obtained by EXAFS analysis of the catalysts Fe01–Fe20 and the $\alpha$- and $\gamma$-$Fe_2O_3$ references; Abs = absorbing atom; Bs = backscattering atom; n(Bs) = number of backscattering atoms; r(Abs-Bs) = distance of absorbing to backscattering atom; $\sigma$ = Debye–Waller-like factor; R = fit index; $E_f$ = Fermi energy; Afac = amplitude reducing factor.

| Catalyst | Abs-Bs | N (Bs) | R (Abs-Bs) [Å] | $\sigma$ [Å$^{-1}$] | |
|---|---|---|---|---|---|
| Fe01 | Fe–O | 4.8 ± 0.24 | 1.944 ± 0.019 | 0.089 ± 0.008 | R = 28.29% |
| | Fe–O | 1.1 ± 0.05 | 2.157 ± 0.021 | 0.045 ± 0.004 | $E_f$ = 3.655 eV |
| | Fe–Fe | 1.0 ± 0.10 | 3.059 ± 0.030 | 0.112 ± 0.011 | Afac = 0.9477 |
| | Fe–Al | 8.1 ± 0.81 | 3.426 ± 0.034 | 0.112 ± 0.011 | |
| | Fe–Fe | 5.0 ± 0.50 | 3.439 ± 0.034 | 0.112 ± 0.011 | |
| Fe025 | Fe–O | 4.6 ± 0.23 | 1.944 ± 0.019 | 0.095 ± 0.009 | R = 38.80% |
| | Fe–O | 1.6 ± 0.08 | 2.169 ± 0.021 | 0.087 ± 0.008 | $E_f$ = 3.137 eV |
| | Fe–Fe | 1.1 ± 0.11 | 3.047 ± 0.030 | 0.107 ± 0.010 | Afac = 0.9072 |
| | Fe–Al | 7.9 ± 0.79 | 3.455 ± 0.034 | 0.112 ± 0.011 | |
| | Fe–Fe | 5.6 ± 0.56 | 3.463 ± 0.034 | 0.112 ± 0.011 | |
| Fe05 | Fe–O | 3.6 ± 0.18 | 1.911 ± 0.019 | 0.087 ± 0.008 | R = 34.13% |
| | Fe–O | 3.1 ± 0.15 | 2.076 ± 0.020 | 0.112 ± 0.011 | $E_f$ = 4.013 eV |
| | Fe–Fe | 0.8 ± 0.08 | 3.063 ± 0.030 | 0.105 ± 0.010 | Afac = 0.8896 |
| | Fe–Al | 7.4 ± 0.74 | 3.397 ± 0.033 | 0.112 ± 0.011 | |
| | Fe–Fe | 4.7 ± 0.47 | 3.414 ± 0.034 | 0.112 ± 0.011 | |
| Fe10 | Fe–O | 3.4 ± 0.17 | 1.907 ± 0.019 | 0.081 ± 0.008 | R = 33.88% |
| | Fe–O | 3.0 ± 0.15 | 2.065 ± 0.020 | 0.112 ± 0.011 | $E_f$ = 2.925 eV |
| | Fe–Fe | 0.5 ± 0.05 | 2.983 ± 0.029 | 0.092 ± 0.009 | Afac = 0.8896 |
| | Fe–Al | 6.2 ± 0.62 | 3.354 ± 0.033 | 0.112 ± 0.011 | |
| | Fe–Fe | 4.0 ± 0.40 | 3.403 ± 0.034 | 0.112 ± 0.011 | |
| Fe20 | Fe–O | 2.5 ± 0.12 | 1.898 ± 0.018 | 0.077 ± 0.007 | R = 33.35% |
| | Fe–O | 3.9 ± 0.19 | 2.025 ± 0.020 | 0.110 ± 0.011 | $E_f$ = 4.384 eV |
| | Fe–Fe | 0.5 ± 0.05 | 2.962 ± 0.029 | 0.081 ± 0.008 | Afac = 0.8217 |
| | Fe–Al | 9.0 ± 0.90 | 3.360 ± 0.033 | 0.112 ± 0.011 | |
| | Fe–Fe | 5.7 ± 0.57 | 3.394 ± 0.033 | 0.112 ± 0.011 | |
| $\alpha$-$Fe_2O_3$ | Fe–O | 3.2 ± 0.16 | 1.961 ± 0.019 | 0.081 ± 0.008 | R = 27.77% |
| | Fe–O | 3.3 ± 0.16 | 2.134 ± 0.021 | 0.110 ± 0.011 | $E_f$ = 2.584 eV |
| | Fe–Fe | 6.3 ± 0.31 | 2.983 ± 0.029 | 0.112 ± 0.011 | Afac = 0.9735 |
| | Fe–Fe | 2.9 ± 0.29 | 3.317 ± 0.033 | 0.112 ± 0.011 | |
| | Fe–Fe | 1.2 ± 0.12 | 3.706 ± 0.037 | 0.063 ± 0.006 | |
| $\gamma$-$Fe_2O_3$ | Fe–O | 0.7 ± 0.03 | 1.868 ± 0.018 | 0.032 ± 0.003 | R = 24.68% |
| | Fe–O | 4.8 ± 0.14 | 2.003 ± 0.020 | 0.105 ± 0.010 | $E_f$ = 3.112 eV |
| | Fe–Fe | 4.4 ± 0.44 | 3.019 ± 0.030 | 0.112 ± 0.011 | Afac = 0.8219 |
| | Fe–Fe | 2.1 ± 0.21 | 3.467 ± 0.034 | 0.087 ± 0.008 | |
| | Fe–Fe | 3.6 ± 0.36 | 5.128 ± 0.051 | 0.112 ± 0.011 | |

Comparison of the first region clearly shows a trend from a coordination of 2.5 oxygen backscatterers at 1.9 Å and 3.9 at 2 Å to the observed iron center of catalyst Fe20, over 3.6 oxygen atoms at 1.9 Å and 3.1 at 2.1 Å at catalyst Fe05, towards a ratio of 4.8 oxygen atoms at 1.9 Å to 1.1 atoms at 2.2 Å for Fe01. If the crystallographic data of the presumed phases of iron [47,48], $\gamma$-$Fe_2O_3$ and $\alpha$-$Fe_2O_3$, are taken into consideration, these numbers can be correlated to tetrahedral and octahedral iron oxide species. Since the averaged Fe-O distance in octahedrons is approx. 2 Å and they are only slightly distorted, which means that the coordination is still somewhat symmetrical, only a contribution of small Fe-O distances from tetrahedrons could lead to a coordination number higher than 3 at 1.94 Å. Thus, the high coordination number of 4.8 oxygen backscatterers of Fe01 can be attributed to a high amount of tetrahedral coordinated iron species. Fe025 seems to be identical, while the coordination changes over 3.6:3.1 for Fe05 to 2.5:3.9 for Fe20, indicating that, here, the ratio of tetrahedral to octahedral iron(III) coordination is decreased, which is in excellent agreement with the information obtained from DRUVS data (Figure 8). The second region, as stated before, contains the nearest iron and aluminum atoms, which can be an indicator for particle size as well as the incorporation of the active site into the support lattice. Unfortunately, no significant differences can be detected in these shells. Accordingly, the short-range order limitation of EXAFS hinders a further structural discussion using these higher-order backscatterers.

### 2.2. Catalytic Activity

The CO oxidation activity of each catalyst was measured both dynamically with a heating ramp of 2 °C/min up to 600 °C (Figure 12), as well as at constant temperatures of 150, 200, and 250 °C (see Figures S30–S34). In the dynamic measurements, the light-off temperatures (defined as temperature where 10% $CO_2$ yield is observed) for Fe01, Fe025, Fe05, Fe10, and Fe20 are 170, 109, 86, 88, and 94 °C, respectively, and 362, 211, 169, 177, and 177 °C for 50% of CO conversion (Table 6). Here, Fe01 shows the lowest activity, followed by Fe025. Fe05, Fe10, and Fe20 only show minor differences in their activity curves, with Fe05 being the most active catalyst.

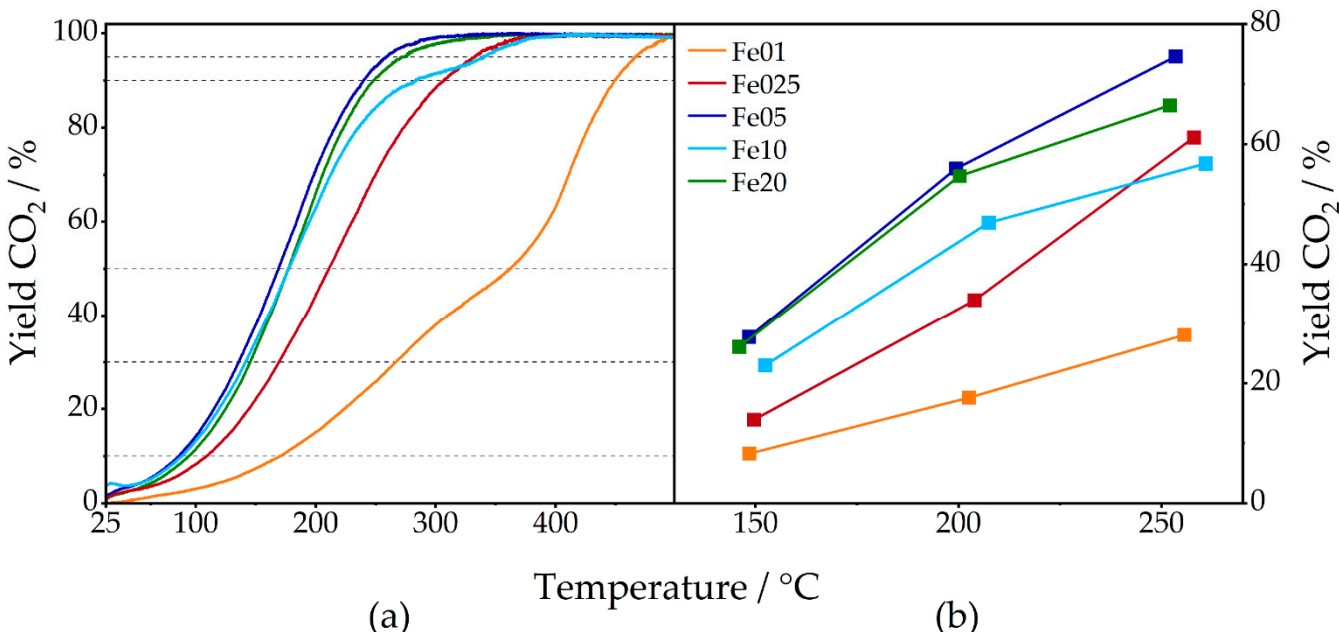

(a)　　　　　　　　　　　　　　(b)

**Figure 12.** Percentual $CO_2$ yield of Fe01 to Fe20 in measurements with continuous heating (**a**) and at constant temperatures (**b**); $CO_2$ yield of 10, 30, 50, 90, and 95% marked with dashed lines; for detailed information, see SI.

**Table 6.** Temperatures of Fe01 to Fe20 corresponding to light-off, 30%, 50%, 90%, and 95% CO conversion, respectively $CO_2$ yield, obtained from measurements with continuous heating. Light-off was defined as the point of 10% CO conversion.

| Catalyst | $T_{Light-off}$ [°C] (10%) | $T_{30}$ [°C] | $T_{50}$ [°C] | $T_{90}$ [°C] | $T_{95}$ [°C] |
|:---:|:---:|:---:|:---:|:---:|:---:|
| Fe01 | 170 | 267 | 362 | 449 | 467 |
| Fe025 | 109 | 169 | 211 | 307 | 332 |
| Fe05 | 86 | 136 | 169 | 240 | 259 |
| Fe10 | 88 | 142 | 177 | 283 | 340 |
| Fe20 | 94 | 146 | 177 | 248 | 272 |

Comparing the temperatures corresponding to 90 and 95% CO conversion, Fe05 is again superior to the other catalysts, followed by Fe20. Fe10 has a lower temperature than Fe025 for 90% conversion, but due to a drop in activity of Fe10, this is inverted for 95% $CO_2$ yield. Turnover frequencies, calculated with total iron loading, are shown in Figure 13.

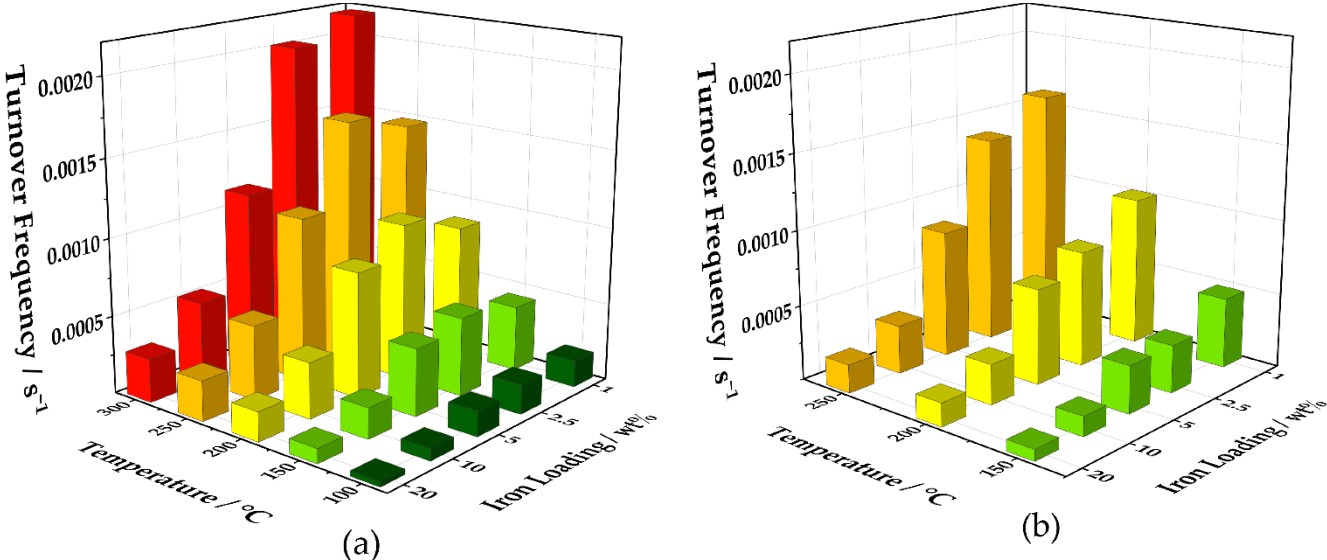

**Figure 13.** Three-dimensional visualization of the turnover frequencies of Fe01 to Fe20 calculated from the values obtained by the measurements with continuous heating (**a**) and at constant temperatures (**b**).

Here, the catalysts with low iron loadings, Fe01, Fe025, and Fe05, are superior to Fe10 and Fe20 at 100, 150, and 200 °C. At higher temperatures, the turnover frequencies of Fe01 and Fe025 are almost twice the TOF of Fe05, while the values for Fe05, Fe10, and Fe20 stagnate. Turnover frequencies obtained from CO oxidation experiments at constant temperatures show a similar trend, where, at 150 °C, Fe01 has a slightly higher TOF than Fe025 and Fe05. Fe10 has a much lower value, followed by Fe20. The turnover frequency of Fe01 doubles 150 to 200 °C and increases with the same amount up to 250 °C, with Fe025 being slightly less active at both temperatures. Fe05 can almost keep up at 200 °C but has a much lower TOF than Fe025 at 250 °C. The values of Fe10 are much lower at all temperatures and Fe20 shows even worse turnover frequencies. It is noteworthy that, for the measurements at constant temperatures, a decay in conversion for each step could be detected after reaching the desired temperature, which stabilized afterwards at 10 to 15% below the initial value (see Figures S30–S34). This decrease derives most certainly from carbon dioxide poisoning of the active sites [65]. Nonetheless, at these measurements at constant temperatures, the superiority of low iron loadings can be visualized even further by comparison of the turnover frequencies (Figure 13b). If the whole amount of iron is taken into the calculation, the TOF of catalyst Fe01 at 250 °C is five-times higher than Fe10 and more than eight-times higher than Fe20.

### 3. Summary and Conclusions

A series of iron oxide catalysts with a range of 1 to 20 wt% iron loading immobilized on a $\gamma$-Al$_2$O$_3$ support was prepared to perform a multidimensional structure–activity correlation. All catalysts contain iron in the oxidation state +3, as confirmed by the isomer shifts and quadrupole splitting of the Mössbauer spectra, as well as the prepeak and edge positions of XANES data. The BET surface area of the catalysts showed only minor changes for Fe01 to Fe10 when compared to the pure $\gamma$-Al$_2$O$_3$ support, whereas the surface area drops from approx. 160 m$^2$/g to 121 m$^2$/g for Fe20. This can be explained by the higher amount of agglomerated and particulate species of Fe20 compared to the other samples, as also indicated by UV/Vis, Mössbauer spectroscopy, as well as the STEM-EDX mapping and PDF analysis.

All characterization methods show smaller particles for the catalysts with lower iron loadings (1–5 wt%) and an increasing agglomeration with loadings of 10 and 20 wt% of iron. DRUVS allows a discrimination of tetrahedrally and octahedrally coordinated iron species resulting in a higher amount of tetrahedrally coordinated centers in catalysts $\leq$5 wt% loading. Furthermore, both types of PDF refinements show that for 5, 10, and 20 wt% iron loading, $\gamma$-Fe$_2$O$_3$ species are present. For 5 and 10 wt%, very small particles of 2–3 nm are formed, and in the case of 20 wt%, significantly larger particles with a size of 8.5 resp. 14 nm (dependent on the refinement method) could be modeled. The results of Mössbauer analysis support these results. Here, Fe01 to Fe05 were assigned to particles less than 13.5 nm, whereas Fe10 and Fe20 do also contain particles above this value. In particular, the superparamagnetism of Fe025 and Fe05 even at 77 K underlines the conjecture of very small particle sizes. Powder XRD analysis indirectly backs these findings, since, for 10 and 20 wt% iron loadings reflexes are visible which can be assigned to $\alpha$- and $\gamma$-Fe$_2$O$_3$, while, for samples $\leq$5 wt%, no bulk iron oxide-related reflexes are detectable—suggesting either poor crystallinity or crystallite sizes too small to be detected by high-resolution XRD. In addition, shifts in the Bragg angle of $\gamma$-Al$_2$O$_3$ reflexes indicate an increase in lattice parameters with higher loadings, which can be assigned to either the incorporation of Fe$^{3+}$ ions into the interfacial area or strain induced at the interface between the support and the catalyst material. The changes in lattice parameters are also visible in the results of PDF analysis.

STEM-EDX mapping of Fe01 to Fe20 also shows large variations in the iron oxide particle sizes. For Fe20, a number of large particles of iron oxide with diameters up to 50 nm can be detected. The amount of these clusters decreases with decreasing iron oxide loading. For Fe01 and Fe025, none of these large clusters are visible. Besides these agglomerates, a fine dispersion of iron oxide species with sizes below the resolution of the microscope can be seen on all observed $\gamma$-Al$_2$O$_3$ particles of Fe01 to Fe20. X-ray absorption spectroscopy of the catalysts was carried out at the iron K-edge. While analysis of the near-edge region does not show any differences throughout the samples, EXAFS analysis also leads to a clear trend of more tetrahedral coordinated iron oxide species with Fe-O distances of approx. 1.94 Å for Fe01 and Fe025, to more Fe-O contributions at distances above 2 Å with increasing iron loading. This can be assigned to a higher amount of octahedral coordinated iron species for Fe10 and Fe20, as also seen in the DRUVS analysis.

The catalysts showed considerable activity in catalytic CO oxidation experiments. Fe05, Fe10, and Fe20 showed high activity at much lower temperatures than Fe01 and Fe025 in the continuous measurements. Over the whole experiment, Fe05 was superior to the other catalysts, followed by Fe20 and Fe10. Fe025 and Fe01 appeared to be least active when comparing the temperatures needed for a particular CO$_2$ conversion yield. However, when converted into turnover frequencies, the catalysts with lower weight loadings of iron were superior to the other catalysts at distinct temperatures, in the continuous as well as the measurements at constant temperatures.

The much higher turnover frequencies of Fe01 and Fe025 can be explained by a higher ratio of more active tetrahedral to less active octahedral iron oxide species, as proven by the deconvolution of the DRUVS data as well as the EXAFS. In addition to this, Fe01 and Fe025 did not show any agglomerates (Fe$_x$O$_y$) in STEM-EDX mapping, while, with increasing iron

loading, large clusters of iron oxide up to 50 nm could be detected. This can be emphasized by the information gathered throughout the Mössbauer analysis as well as the DRUVS data, which both showed large contributions of iron oxide oligomers and $Fe_xO_y$ particles. This means that an increasing amount of iron oxide is present in the bulk phase with increasing iron loading, which is likely to be less accessible for catalytic purposes. Nevertheless, Fe05 to Fe20 showed conversion at much lower temperatures than Fe01 and Fe025. While the ratio of tetrahedral to octahedral iron oxide species of Fe10 and Fe20 is lower than that of Fe01 and Fe025, the absolute amount of tetrahedrally coordinated $Fe^{3+}$ still increases with increasing iron loading. The fact that Fe05 showed the highest activity by means of the lowest temperatures needed leads to the assumption that Fe05 is the best compromise between a high amount of more active tetrahedral iron oxide species and the amount of iron oxide present in the less or non-active bulk phase. The latter is higher for Fe10 and Fe20, which is most probably the reason that they are not better than Fe05.

Taking all these results into consideration, Fe05 presents an excellent starting point for the optimization of iron-based CO oxidation catalysts. An iron oxide catalyst with a high amount of iron present in a tetrahedral coordination geometry that is finely dispersed and easily accessible for catalysis without any agglomeration, i.e., without iron oxide in the bulk phase, could exhibit competitive catalytic activity in CO oxidation. Since the process of agglomeration is a problem most likely occurring during the annealing process and could probably even be linked to the formation of higher amounts of $\alpha$-$Fe_2O_3$, or solely octahedrally coordinated iron oxide species, a synthesis route towards this aim needs to be developed. Potential solutions comprise either the alteration of the heating process itself, by variation of the temperature ramp applied or a stepwise temperature program, or the successive annealing of small amounts of iron precursor impregnated on the support. Such a multistep impregnation–calcination process would benefit from the fact that iron oxide is less mobile than the non-annealed $Fe(acac)_3$ precursor, potentially hindering agglomeration on the support surface.

## 4. Experimental Methodology

### 4.1. Preparation of $\gamma$-$Al_2O_3$

The $\gamma$-$Al_2O_3$ support was synthesized by calcination of PuralBT® (Sasol Germany GmbH) in a furnace under atmospheric air. Heating with a ramp of 5 °C/min was carried out up to 600 °C, where the sample was further calcined for 3 h. Phase purity was checked by X-ray diffraction measurements in a 2θ range of 15 to 80 degrees (see Figure S1).

### 4.2. Preparation of Catalysts Fe01 to Fe20

A 0.25 M solution of Fe(III) acetylacetonate in a mixture of N-Methyl-2-pyrrolidone and tetrahydrofuran (1:1) was prepared and added to a suspension of previously synthesized $\gamma$-$Al_2O_3$ in tetrahydrofuran. Details are given in Table 7. After stirring for 30 min, the mixture was heated to 140 °C and the solvents were slowly removed under reduced pressure. The dry reddish powders were then annealed in a furnace under air with a heating ramp of 5 °C/min up to 600 °C and held at this temperature for 3 h. The reported weight loadings of iron refer to the mass of iron inserted during synthesis. The amounts of iron and $\gamma$-$Al_2O_3$ were calculated to add up to 100 wt%.

**Table 7.** Parameters for the catalyst preparation of Fe01 to Fe20.

| Catalyst | Weight Loading [wt%] | m (Fe) [mg] | n (Fe(acac)₃) [mmol] | m ($\gamma$-Al₂O₃) [g] |
|---|---|---|---|---|
| Fe01 | 1 | 20 | 0.358 | 1.98 |
| Fe025 | 2.5 | 50 | 0.895 | 1.95 |
| Fe05 | 5 | 100 | 1.79 | 1.90 |
| Fe10 | 10 | 200 | 3.58 | 1.80 |
| Fe20 | 20 | 400 | 7.16 | 1.60 |

### 4.3. Catalytic Experiments

For catalytic experiments the samples were homogenized in a mortar, pressed to a pellet and then granulated and sieved to a fraction of 125 to 250 μm. Then, 333 mg of the catalyst was filled in a quartz glass tube with an inner diameter of 8 mm. The filling was plugged on both sides with quartz wool and the reactor was mounted into a clamshell oven. K-type thermocouples were inserted from both sides to log the exact temperatures at the gas entrance and exit of the packed bed. Continuous measurements were carried out during heating with a rate of 2 °C/min up to 600 °C with a permanent gas feed of 1000 ppm CO and 10 vol% $O_2$ in inert gas balance to a total flow of 500 mL/min. For activation, the catalysts were heated under a constant argon flow of 500 mL/min up to 600 °C with a heating rate of 5 °C/min and cooled down again to room temperature. For good comparison, these parameters were chosen according to previous work [43] and with exclusion of diffusion limitation. For experiments at constant temperatures, the catalysts were cooled down in inert gas after the continuous measurement to 150 °C. Then, the above-mentioned gas feed was applied and the catalytic activity measured at 150, 200, and 250 °C, each for 2 h. Further details about the experiment can be found in the Supplementary Information.

### 4.4. Analytics

The surface areas were calculated via the BET method from nitrogen physisorption, which was carried out at 77 K using a Quantachrome Autosorb 6 after degassing the samples for 12 h at 120 °C.

High-resolution powder X-ray diffractometry was carried out at beamline P24 at DESY (Hamburg/Germany) with an incident radiation of 20 keV ($\lambda$ = 0.619 Å) and a MarCCD 165 detector. For processing of the obtained Bragg circles, i.e., the transformation into a diffractogram, the free software Datasqueeze [66] was used. For calibration of the spectra, lanthanum hexaboride was used as a reference.

[57]Fe Mössbauer spectra of Fe025 to Fe20 at room temperature and of Fe025 and Fe10 at 77 K were obtained in transmission geometry using a constant acceleration spectrometer (WissEl GmbH, Mömbris, Germany) with a 512-channel analyzer and a [57]Co source implemented in a Rh matrix. A continuous flow cryostat (OptistatDN, Oxford Instruments, Abingdon, UK) was utilized to perform experiments at 77 K with an accuracy of $\pm 1.0$ K. Transmitted radiation was measured by a proportional counter. Calibration was performed with $\alpha$-Fe. For further analysis, spectroscopic data were transferred from the multi-channel analyzer to a PC. The spectra were analyzed, employing the public domain program Vinda [67] running on an Excel 2003® platform, by least-squares fits using Lorentzian line shapes. In addition to the Mössbauer parameters isomer shift $\delta$, quadrupole splitting $\Delta E_Q$, and the line widths at half maximum $\Gamma$, the spectral area and area ratios of the components to each other were determined.

[57]Fe Mössbauer spectra of Fe01 at room temperature and Fe05 and Fe20 at 77 K were taken on a different spectrometer (WissEl GmbH), and temperature was controlled by an MBBC-HE0106 Mössbauer He/$N_2$ cryostat within an accuracy level of $\pm 0.3$ K. The collected spectra were analyzed by the WinNormos [68] software using a least-square fitting procedure assuming Lorentzian peak shapes. The obtained correlation coefficients were always above 0.95, indicating appropriate accuracy of the deconvolution.

Diffuse reflectance UV/Vis spectroscopy was carried out at a Lambda 18 (Perkin Elmer, Waltham, MA, USA) in the range of 200 to 900 nm. For background correction, a Lorentz-type function was fitted to the regions of 200 to 215 nm and 700 to 900 nm and then subtracted from the spectrum. Deconvolution was carried out with the NLFit function of Origin 2020b [69] via manual selection of 5 (respectively, 6) initial wavelengths as starting points for the fit. Lower limits were set for Peak 1 at 260 nm, Peak 3 at 340 nm, and $Y_0$ according to the lowest point of the spectrum to keep the fit within these boundaries.

Scanning transmission electron microscopy (STEM) images were obtained using the probe-side Cs-corrected JEOL JEM-ARM200F, equipped with a cold field emission gun and a JEOL SDD detector for the acquisition of energy-dispersive X-ray spectra (EDS).

TEM specimens of all catalysts were prepared by dispersing the synthesis products in isopropyl alcohol and placing a droplet of the dispersion on a Lacey grid. The analysis of all specimens was conducted with a high tension of 200 kV, a semi-convergence angle of 25 mrad and a maximum image resolution of 70 pm owing to the Cs correction. High angle annular dark-field (HAADF) images were acquired on an annular dark-field detector with a collection angle ranging from $51 \pm 2$ to $180 \pm 2$ mrad at a camera length of 12 cm. At these settings, the intensity contrasts in HAADF images can be explained within the well-known Z-contrast model by Pennycook [70], in which the contrast is proportional to the atomic number $Z^{\sim 2-x}$ (x between 0.3 and 0.7, accounting for inelastically scattered electrons) [71]. Dwell time and image resolution of EDX mappings were chosen to keep specimen drift during acquisition as small as possible. Elemental maps of iron, aluminum, and oxygen were obtained using K-edges.

X-ray absorption spectroscopy (XAS) experiments at the Fe K-edge (7112 eV) were performed at PETRA III beamline P65 at Deutsches Elektronen-Synchrotron DESY (Hamburg/Germany) using a Si(111) double crystal monochromator at a maximum beam current of 100 mA. Energy calibration of the monochromator was controlled using a Fe foil. The samples were diluted with cellulose (Sigma-Aldrich, St. Louis, MO, USA), homogenized in a mortar and then pressed to a pellet. Fe01 was measured in fluorescence mode using a passivated implanted planar silicon (PIPS) detector. Fe025 to Fe20 were measured in transition mode using ionization chambers in front of and behind the sample. The spectra were measured in step scan mode, which means that the spectra are divided into regions with different step sizes (Table 8).

**Table 8.** Step scan parameters used in X-ray absorption spectroscopy.

| Energy [eV] | Step Size | Time per Point [s] |
|---|---|---|
| 6962–7062 | 5 eV | 0.2 |
| 7062–7092 | 3 eV | 0.2 |
| 7092–7142 | 0.5 eV | 0.2 |
| 7142–8112 | 0.5 Å$^{-1}$ | 0.2 |

Absorption edge energy ($E_0$) was defined as the center of the jump height. For EXAFS analysis, the background was subtracted from the obtained spectra as a Victoreen-type polynomial [72,73], followed by determination of the smooth part of the spectrum by a piecewise polynomial, optimized to yield minimal low-R components for the resulting Fourier transformation. After division by the smoothed part, the photon energy was transformed into the photoelectron wavenumber $k$. $\chi(k)$ was weighted with $k^3$ for the fitting with the program EXCURV98 [74], which utilizes the EXAFS equation (Equation (1)) in the form of pseudo-radial distribution functions.

$$\chi(k) = \sum_j s_0^2(k) \frac{N_j}{k r_j^2} F_j(k) e^{-2\sigma_j^2 k^2} e^{\frac{2r_j}{\lambda(k)}} \sin\left[2kr_j + \varphi_{ij}(k)\right] \tag{1}$$

Here, inelastic effects are represented by the amplitude reducing factor $s_0^2(k)$ and $\lambda$, the mean free path length, while the number of backscattering atoms $N_j$, their distance to the observed atom $r_j$ and the Debye–Waller-like factor $\sigma^2$ take structural parameters into account.

Synchrotron XRD data for PDF analysis were acquired at I15-1 beamline on a Diamond Light Source with an X-ray energy of 65.4 keV. At the beamline, all samples were measured in Kapton® capillaries with a diameter of 1 mm for 2 min. The powder diffraction patterns were collected using a Perkin Elmer XRD 4343 CT detector, resulting in a Q-range of 0.5–34.5 Å$^{-1}$. Radial integration was done with the DAWN software [75].

Powder X-ray diffraction measurements for PDF analysis were carried out at room temperature with a STOE STADI P Mython2 4 K diffractometer (Ge(111) monochromator; Ag K$_{\alpha 1}$ radiation, $\lambda$ = 0.5594 Å) using four Dectris MYTHEN2 R 1K detectors in Debye–

Scherrer geometry. Samples were measured in 1-mm-diameter Kapton® capillaries for 12 h. The Q-range was 20.4 Å$^{-1}$ [76]. PDF calculation of laboratory data was carried out with xPDFsuite [77].

PDF refinements were carried out with diffpy-cmi [78]. The γ-$Al_2O_3$ support was refined against a cubic (spinel) Fd$\overline{3}$m crystal structure [50], containing $Al^{3+}$ ions on tetrahedral and octahedral spinel positions, as well as additional $Al^{3+}$ ions on non-spinel positions. Refinable variables were the occupancy of these positions (8a, 16c, 16d, 48f Wyckoff), thermal parameters $B_{iso}$ of the $Al^{3+}$ ions, the lattice parameter *a*, spherical particle size accounting for small crystalline domains in the porous support, the phase scale factor and $\delta_2$ accounting for correlated motion of nearest neighbor atoms, described by Jeong et al. [79].

The PDF of the loaded sample was refined with a γ-$Al_2O_3$ support as one phase, where all occupancies and thermal parameters were constrained to the values from the previously refined γ-$Al_2O_3$. Lattice parameter, scale, spherical particle size and $\delta_2$ were refined accounting for the structural changes evolving with the introduction of the $Fe_2O_3$ on γ-$Al_2O_3$. γ-$Fe_2O_3$ was added as a second phase to the fit and refined against a cubic P4$_3$32 crystal structure [80], refining its lattice parameter *a*, the thermal parameters $B_{iso}$, the spherical particle size, the scale and the correlated motion factor $\delta_2$. The d-PDFs were refined against a γ-$Fe_2O_3$ phase in the manner mentioned above, adding a second cluster phase, in which the spherical particle size was constrained to 1 nm. For all refinements, laboratory PDF data were used.

**Supplementary Materials:** The following are available online at https://www.mdpi.com/article/10.3390/catal12060675/s1, Figure S1 Experimental powder X-ray diffractogram of the as prepared γ-$Al_2O_3$ support compared to a calculated powder pattern, obtained via the program Mercury [81] from a single-crystal structure [49,50]. Figure S2 Powder X-ray diffractograms of Fe01 to Fe20, compared to the γ-$Al_2O_3$ support, obtained at the in-house setup. Figure S3 XRD of 20% Fe loading (red) together with unloaded $Al_2O_3$ support (blue) and difference curve (grey, in offset). The top panel shows the direct subtraction of experimental data resulting in strong dips at Q values where strong $Al_2O_3$ reflexes of unloaded support appear, because the modified lattice parameters of the $Al_2O_3$ are not taken into account. Bottom panel contains the $Al_2O_3$ pattern compressed by the compressing factor s for a better match of the $Al_2O_3$ reflexes, thus creating less strong dips in the difference curve, which is indexed with γ-$Fe_2O_3$. Figure S4 PDF refinement of the γ-$Al_2O_3$ support over the range 1–80 Å (top) and magnification of the 1–10 Å range (bottom). Figure S5 Refinement of only the short-range order of the γ-$Al_2O_3$ support between 1–10 Å with different occupation of octahedral and tetrahedral positions. Table S1 Refinement values. Figure S6 Mössbauer spectrum of Fe025 (black dots) obtained at 77 K and corresponding fit of the doublet (red). Table S2 Parameters obtained by Mössbauer spectroscopy of Fe025 at 77 K. Figure S7 Mössbauer spectrum of Fe05 (black dots) obtained at 77 K and corresponding fit of the doublet (red). Table S3 Parameters obtained by Mössbauer spectroscopy of Fe05 at 77 K. Figure S8 Mössbauer spectrum of Fe10 (black dots) obtained at 77 K, fit of the doublet (red), sextet (blue) and the cumulative fit (green). Table S4 Parameters obtained by Mössbauer spectroscopy of Fe10 at 77 K. Figure S9 Mössbauer spectrum of Fe20 (black dots) obtained at 77 K, fit of the doublet (red), sextet (blue) and the cumulative fit (green). Table S5 Parameters obtained by Mössbauer spectroscopy of Fe20 at 77 K. Figure S10 DRUV spectrum and deconvolution of Fe01. Table S6 Parameters of the peak deconvolution of Fe01. Table S7 Assignment of the peaks and their normalized areas to tetrahedral, octahedral and oligomerized iron oxide. Percentual amounts of the respective iron species of the overall amount of catalyst, calculated from the percentual areas multiplied by the iron loading. Figure S11 DRUV spectrum and deconvolution of Fe025. Table S8 Parameters of the peak deconvolution of Fe025. Table S9 Assignment of the peaks and their normalized areas to tetrahedral, octahedral and oligomerized iron oxide. Percentual amounts of the respective iron species of the overall amount of catalyst, calculated from the percentual areas multiplied by the iron loading. Figure S12 DRUV spectrum and deconvolution of Fe05. Table S10 Parameters of the peak deconvolution of Fe05. Table S11 Assignment of the peaks and their normalized areas to tetrahedral, octahedral and oligomerized iron oxide. Percentual amounts of the respective iron species of the overall amount of catalyst, calculated from the percentual areas multiplied by the iron loading. Figure S13 DRUV spectrum and deconvolution of Fe10. Table S12 Parameters of the peak deconvolution of Fe10. Table S13

Assignment of the peaks and their normalized areas to tetrahedral, octahedral and oligomerized iron oxide. Percentual amounts of the respective iron species of the overall amount of catalyst, calculated from the percentual areas multiplied by the iron loading. Figure S14 DRUV spectrum and deconvolution of Fe20. Table S14 Parameters of the peak deconvolution of Fe20. Table S15 Assignment of the peaks and their normalized areas to tetrahedral, octahedral and oligomerized iron oxide. Percentual amounts of the respective iron species of the overall amount of catalyst, calculated from the percentual areas multiplied by the iron loading. Figure S15 Prepeak area of Fe01 and the background to remove the main edge, obtained by a Boltzmann function. Table S16 Fit parameters of the background fit for Fe01. Figure S16 Prepeak area of Fe01 with inverse polynomial fit after removal of the main edge. Table S17 Fit parameters of the prepeak fit for Fe01. Figure S17 Prepeak area of Fe025 and the background to remove the main edge, obtained by a Boltzmann function. Table S18 Fit parameters of the background fit for Fe025. Figure S18 Prepeak area of Fe025 with inverse polynomial fit after removal of the main edge. Table S19 Fit parameters of the prepeak fit for Fe025. Figure S19 Prepeak area of Fe05 and the background to remove the main edge, obtained by a Boltzmann function. Table S20 Fit parameters of the background fit for Fe05. Figure S20 Prepeak area of Fe05 with inverse polynomial fit after removal of the main edge. Table S21 Fit parameters of the prepeak fit for Fe05. Figure S21 Prepeak area of Fe10 and the background to remove the main edge, obtained by a Boltzmann function. Table S22 Fit parameters of the background fit for Fe10. Figure S22 Prepeak area of Fe010 with inverse polynomial fit after removal of the main edge. Table S23 Fit parameters of the prepeak fit for Fe10. Figure S23 Prepeak area of Fe20 and the background to remove the main edge, obtained by a Boltzmann function. Table S24 Fit parameters of the background fit for Fe20. Figure S24 Prepeak area of Fe20 with inverse polynomial fit after removal of the main edge. Table S25 Fit parameters of the prepeak fit for Fe20. Figure S25 Prepeak area of $\alpha$-$Fe_2O_3$ at the Fe K-edge and the background to remove the main edge, obtained by a Lorentzian-type function. Table S26 Fit parameters of the background fit for $\alpha$-$Fe_2O_3$. Figure S26 Background-corrected prepeak of $\alpha$-$Fe_2O_3$ and corresponding Gaussian-type peak fits and cumulative peak fit. Table S27 Fit parameters of the prepeak fit of $\alpha$-$Fe_2O_3$. Figure S27 Prepeak area of $\gamma$-$Fe_2O_3$ at the Fe K-edge and the background to remove the main edge, obtained by a Lorentzian-type function. Table S28 Fit parameters of the background fit for $\gamma$-$Fe_2O_3$. Figure S28 Background-corrected pre-peak of $\gamma$-$Fe_2O_3$ and corresponding Gaussian-type peak fits and cumulative peak fit. Table S29 Fit parameters of the prepeak fit of $\gamma$-$Fe_2O_3$. Table S30 Prepeak and edge positions of Fe01 to Fe20 and the $\alpha$- and $\gamma$-$Fe_2O_3$ references. Figure S29 $k^3\chi(k)$ of the EXAFS spectra (left) and the corresponding Fourier-transformed functions (right) of catalyst Fe01 to Fe20 and the fitted spectra. Figure S30 CO oxidation experiment of Fe01 at constant temperatures. Figure S31 CO oxidation experiment of Fe025 at constant temperatures. Figure S32 CO oxidation experiment of Fe05 at constant temperatures. Figure S33 CO oxidation experiment of Fe10 at constant temperatures. Figure S34 CO oxidation experiment of Fe20 at constant temperatures.

**Author Contributions:** Conceptualization, methodology, preparation of the catalysts, processing and analysis of PXRD, BET, HRPXRD (P24 data), DRUVS and XAS data, collection of XAS data, catalytic experiments and interpretation, writing—original draft preparation, S.S.; data acquisition of XRD (I-15 data) and PDF, processing and analysis of XRD (I-15) and PDF, interpretation and discussion of the XRD (I-15) and PDF data, writing—PDF part of the manuscript, N.P. and M.Z.; writing—review and editing of XRD and PDF part, N.P. and M.Z.; acquisition, processing and interpretation of STEM-EDX data, J.B.; writing—review and editing of STEM-EDX part, J.B. and J.K.N.L.; acquisition, processing and interpretation of Mössbauer spectra, A.O., V.S., C.S. and S.K.; writing—review and editing of Mössbauer part, A.O., V.S., C.S. and S.K.; writing—review and editing, R.S.; writing—review and editing, supervision, M.B.; resources, M.Z., J.K.N.L., V.S., S.K. and M.B.; funding acquisition, M.Z., S.K. and M.B. All authors have read and agreed to the published version of the manuscript.

**Funding:** This research was funded by the German Research Foundation DFG, grant number BA 4467/5-1 and KU 1459/6-1, and within the framework of the SPP2080, grant number ZO 369/2-1 and BA 4467/8-1. J.B. received funding within the NRW Forschungskolleg "Leicht-Effizient-Mobil".

**Data Availability Statement:** Data are available from the corresponding author upon reasonable request.

**Acknowledgments:** We would like to acknowledge DESY for granting beamtime for XAS at beamline P65 (Edmund Welter) and for HRPXRD at beamline P24 (local contact), Johannes Bitzer & Wolfgang Kleist for measurement of HRPXRD and Annelies DeCuyper for measurement of the DRUVS spectra,

Andrej Paul for in-house XRD measurements and Markus Schmitz for BET measurements. Further, we acknowledge Diamond Lightsource for beamtime at I-15 with local support by Phil Chater.

**Conflicts of Interest:** The authors declare no conflict of interest.

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
