# Peer review of "Quality or Quantity? How Structural Parameters Affect Catalytic Activity of Iron Oxides for CO Oxidation"

_catalysts, doi:10.3390/catal12060675_

Round 1

Reviewer 1 Report

The authors reported the activity of iron oxides for CO oxidation. The results sound good. However, the paper can be improved before accepted.

The iron oxides NPs should be characterzied using TEM. The size is very important.

The actvie sites should be determined using TPD.

The reaction mechanism should be discussed. The related paper can be referenced, such as J. Chem. Phys. 2011,134, 034305. 

Author Response

Dear sir or madam,

thank you very much for your kind and very helpful remarks.

  1. We will try to address the particle size distribution from our TEM-EDX mappings. However, due to time restrictions at the microscope the number of points investigated was not that large to have a statistically relevant distribution of particle sizes. Due to that reason, we tried to address the question of particle sizes with other analytical methods such as UV/Vis and XAS.
  2. TPD is a method we also contemplated for investigation of the iron oxide active sites. Unfortunately, with standard experimental conditions, a differentiation between iron and aluminum sites is not possible, diminishing the significance of this method. We have tested various methods with NH3, CO2, CO or even HCN as a probe molecule, but unfortunately none of them helped to distinguish between active sites and support or lead to additional results that were beneficial for our structure-activity-correlation.
  3. Thank you for this excellent remark. The reaction mechanism of CO oxidation on these iron oxide catalysts is still unclear but absolutely crucial in order to understand their working principles. However, we do not think that with the herein discussed results we have a solid basis to discuss the reaction mechanism. We would need to carry out kinetic experiments to do so, which was not the intention of this work. However, we do absolutely agree that this would be the next step to deepen our knowledge in this topic.

Reviewer 2 Report

Main question addressed by the research: The work addresses the issues related to  How Structural Parameters Affect Catalytic Activity of Iron Oxides for CO Oxidation.
Originality and relevance of the topic: The topic is relevant to the field and it considers a suitable research gap.
Added value of the paper:  The manuscript takes into account the study based on a systematic structure-activity-correlation, however the main purpose of it is not clearly stated. The paper should include main goal and direct applications at the end of the introduction.

Quality of figures: Very good and clear
Specific improvements for the paper to be considered:

  1. Abstract is too short and general. It should summarize the main findings and applications of the paper. 
  2. Introduction is a bit too short.  More discussion and state of the art for direct applications should be included. Additionally the context of the research should be set in a clearer perspective.
  3. There is a big issue with structure in this paper. Materials and methods should go before Results or then it is difficult to follow what was done.
  4. Figure 12b only considers three experimental points. Why is that? Tendency is not justified just with three points.
  5. The selection of the optimal catalytic conditions is unclear, please add more results in these section.
  6. The conclusions are poor and they would need more elaboration so they clearly match the results. It is weird to find a Figure in the conclusions. Conclusions should be able to summarize the results, not to show them again.

Author Response

Dear sir or madam,

thank you very much for your kind and very helpful remarks.

In the revised version we addressed all of them. Especially we tried to specify the main goal and application of the work in the introduction as well as to describe the conclusion more precisely.

  1. Thank you for this remark. We totally agree and added a summary of the main findings as well as a short conclusion to the abstract.
  2. We are not sure whether we understand this remark correctly. In our introduction we tried to describe the overall aim of this research field, which is the usage of such catalysts in automotive or stationary applications for reduction of toxic emissions. We discussed the need for such catalysts, the state-of-the-art noble metal catalysts and why a replacement with abundant alternatives is needed. Then we tried to describe the most relevant research on iron regarding this topic as well as its drawbacks, hence reasoning our work described in the publication. To put this in a better perspective, we specified the main goal and the targeted application of our work further. In our opinion the given background and classification in current ambitions to exchange the commonly used noble metals with iron are comprehensive.
  3. We absolutely agree that the order of figures and tables and their discussion should follow a strict rule. We tried to put and refer everything in chronological order but wanted to place figures and tables in a way that there is not that much space wasted. However, we think that the placement of figures and charts is an editorial task and will be carried out by the board of editors according to the journal standards.
  4. Additional to the measurements under continuous heating we wanted to investigate the catalytic activities at constant temperatures. Therefore, we chose three temperatures, 150, 200 and 250 °C. These temperatures were chosen because they are all in the temperature range, where all catalyst show catalytic activity but are still below full conversion. This leads to a good comparability which is in excellent agreement to the results from measurements under continuous heating. Since we did not put a regression into these data points, we think that three data points are sufficient for a comparison of their catalytic activity.
  5. We thank the referee for this excellent remark. We chose the experimental parameters according to previous work of our group for a better comparison and also assured, that there was no diffusion limitation in our experiments. We added this information in the experimental part of the publication.
  6. In the revised article we tried to address the remark made by the referee by specification of our conclusion, which also rendered the figure redundant.

We hope that the aim of the publication, its application and especially the discussion of the results, respectively the outcome of our work, is now clearer and we thank the referee again for the fast response and excellent remarks.

Reviewer 3 Report

I would recommend the acceptance of this article in its current format. 

This is overall a nice work discussing an important topic regarding the CO oxidation activity relation with respect to structural details, which will be of great interest to the readership of Catalysts. The characterizations including XRD, PDF, XAS, etc. of the Iron oxides are comprehensive, and the conclusions are supported by the experimental evidence. 

Author Response

Dear sir or madam,

thank you very much for your fast and very kind response to our work. It is very much appreciated.

Round 2

Reviewer 1 Report

The author didn't revise their manuscript well according to the reviewer's suggestions. The result of the experiment is not good, but some results should also be provided in the revised manuscript or response letter.

Author Response

Dear Sir or Madame,

attached you will find some experimental data we obtained during the TPD analysis of catalysts, which are very related to the presented in our manuskript (in terms of composition and loading). 

Concerning your comments on our revised version, we did our best to either implement your suggestions in the manuskript or to explain our reasons why we did not follow your concerns. If there are points left, which we can improve, please name them more detailed, so we can address them.

Reviewer 2 Report

Paper should be published.

Author Response

Thank you very much.